# The Effectiveness of the Schroth Best Practice Program and Chêneau-Type Brace Treatment in Adolescent Idiopathic Scoliosis: Long-Term Follow-Up Evaluation Results

**DOI:** 10.3390/children10020386

**Published:** 2023-02-16

**Authors:** Tuğba Kuru Çolak, Burçin Akçay, Adnan Apti, İlker Çolak

**Affiliations:** 1Faculty of Health Sciences, Department of Physiotherapy and Rehabilitation, Marmara University, 34854 Istanbul, Turkey; 2Faculty of Health Sciences, Department of Physiotherapy and Rehabilitation, Bandırma Onyedi Eylül University, 10200 Bandirma, Turkey; 3Faculty of Health Sciences, Department of Physiotherapy and Rehabilitation, Istanbul Kültür University, 34191 Istanbul, Turkey; 4Department of Orthopaedics and Traumatology, VM Medical Park Maltepe Hospital, 34846 Istanbul, Turkey

**Keywords:** adolescents, brace, exercise, scoliosis

## Abstract

Background: Although the number of studies showing the efficacy of conservative treatment in adolescent idiopathic scoliosis has increased, studies with long-term follow-up are very limited. The aim of this study was to present the long-term effects of a conservative management method including exercise and brace in adolescent idiopathic scoliosis patients. Methods: This retrospective cohort study included patients with idiopathic scoliosis who presented at our department and were followed up for at least 2 years after completing the treatment. The main outcome measurements were the Cobb angle and angle of trunk rotation (ATR). Results: The cohort participants were 90.4% female, with a mean age of 11 years and the maximum Cobb angle was mean 32.1°. The mean post-treatment follow-up period was 27.8 months (range 24–71 months). The improvements after treatment in mean maximum Cobb angle (*p* < 0.001) and ATR (*p* = 0.001) were statistically significant. At the end of treatment, the maximum Cobb angle was improved in 88.1% of the patients and worsened in 11.9% compared to baseline. In the long-term follow-up evaluations, 83.3% of the curvatures remained stable. Conclusions: The results of this study showed that moderate idiopathic scoliosis in growing adolescents can be successfully halted with appropriate conservative treatment and that long-term improvement is largely maintained.

## 1. Introduction

Scoliosis is a complex deformity of the spine and trunk with various etiologies. However, adolescent idiopathic scoliosis (AIS), the most prevalent form, has an undetermined etiology and affects 2–3% of the adolescent population [1,2,3].

Adolescents with mild scoliosis typically do not exhibit any symptoms other than a decline in self-esteem and a negative perception of themselves [4]. However, as the curvatures tend to progress, patients are likely to experience health and social issues in adulthood when the curvatures exceed 30° during skeletal maturity [5,6].

When the spinal curvatures exceed 40–50°, surgical treatment is recommended to stop the progression of the curvature [7]. Lonstein and Carlson reported that untreated idiopathic scoliosis < 30° tends to progress during the growth period and even after skeletal maturity [6]. Therefore, it is important that the appropriate treatment is provided at the right time.

Current treatment options are scoliosis-specific physiotherapy methods, corrective bracing, or surgery. Scoliosis-specific physiotherapeutic exercises aim to address the spinal curvature itself, attempting to achieve self-correction with active trunk movements under visual and proprioceptive control. Previous studies have shown that idiopathic scoliosis in adolescents can be successfully treated using a full-time brace for several months or until skeletal maturity, and is able to prevent progression of the deformity and avoid the need for surgical treatment [7,8,9].

When applied together with the use of a brace, scoliosis-specific exercises are also very important, as they help strengthen the muscles on either side of the spine, preventing them from becoming atrophied through decreased use and increased hypokyphosis when the brace is applied [9].

A number of well-designed studies including conservative treatment of adolescent idiopathic scoliosis have been conducted and the evidence has become stronger in recent years [10,11,12,13]. However, there is a lack of information on the long-term results of patients who have received conservative treatment. The aim of this study was to determine the long-term effects of a conservative management (including exercise and brace) method on Cobb angle and angle of trunk rotation in adolescent patients.

## 2. Materials and Methods

The study included adolescent idiopathic scoliosis patients who presented to our department, between 2010 and 2022. All the study procedures were performed in accordance with the Declaration of Helsinki and approval for the study was granted by the Ethics Committee ofBandırma Onyedi Eylül University.

The inclusion criteria were defined as a diagnosis of adolescent idiopathic scoliosis, age ≥ 10 years, Cobb angle ≥ 20°, no other treatment which might affect scoliosis, and follow-up of at least 24 months after the end of treatment. The exclusion criteria were the presence of any contraindications to exercise or brace treatment, accompanying mental or psychological problems, any chronic neurological–muscular or rheumatic diseases, any previous or ongoing treatment, accompanying orthopedic problems, non-idiopathic scoliosis, or refusal of the recommended treatment.

The age of menarche and time since menarche were determined in the initial examination. A detailed physical and radiological examination was performed by an experienced physician to exclude the diagnosis of secondary scoliosis. Height and weight were assessed with the same stadiometer at least every 6 months.

The Cobb method was used to measure the degree of scoliosis [14]. Changes in the sagittal plane of the spine were examined on the lateral radiographs. Lateral radiography was not routinely prompted after the initial evaluation, but repeated lateral radiographs were ordered if needed. Sagittal plane evaluation was performed on lateral radiographs with the Cobb method.

To evaluate the effectiveness of the brace, 6 weeks after fitting each new brace, an in-brace radiographic assessment was performed. To reduce radiation exposure, if there was no increase in ATR measurements, routine X-rays were taken without the brace every 12 months. Before the radiographic assessments, patients did not wear the brace for at least 24 h, and then the X-rays were taken. After the end of treatment, patients were called for radiological and clinical evaluation at 1-year intervals.

The difference between the Cobb angle in the follow-up examinations and before treatment was calculated. Based on this analysis, according to the SOSORT (International Society on Scoliosis Orthopaedic and Rehabilitation Treatment) guidelines, three possible outcomes were categorized: curve correction (>−5°), curve stabilization (>−5° and <5°), and curve progression (>5°) [15].

The topographic features and pattern of scoliosis were determined on the first X-ray. The Augmented Lehnert-Schroth (ALS) classification was used for pattern classification to determine the type of curvature. According to the ALS classification, the curvatures were classified as 3CH (functional 3-curve with hip prominence), 3CTL (functional 3-curve, thoracolumbar with hip prominence), 3CN (functional 3-curve, neutral with balanced pelvis), 3CL (functional 3-curve with long lumbar counter curve), 4C (functional 4-curve, double major), 4CL (functional 4-curve with single lumbar), and 4CTL (functional 4-curve with single thoracolumbar) [16,17]. The functional three-curve patterns define primarily thoracic curves, and the functional four-curve patterns define double major or single lumbar and thoracolumbar curves with additional lumbosacral curves [16,17].

The Risser sign was evaluated on the anteroposterior radiograph [18,19], and the Tanner stage assessment was performed for each patient [20,21].

The angle of trunk rotation was assessed with a Bunnell Scoliometer™ and the readings were obtained with the patient standing in a forward bending position and the maximum ATR measured was recorded [22].

All the patients received the Schroth Best Practice program plus brace. The Schroth Best Practice approach is the most up-to-date version of the Schroth method developed by Schroth’s grandson Dr. Weiss in the light of evidence-based information. The original Schroth method was developed in the 1920s for thoracic curves > 70°. The method was updated by Schroth’s daughter, the physiotherapist Lehnert-Schroth in the 1960s. Finally, it was adapted by Dr. Weiss to become the method currently known as the Schroth Best Practice method. Based on the original method, Dr. Weiss created a holistic treatment program including the exercise patterns applicable to smaller and middle curves, added sagittal plane corrections, activities of daily living, and bracing techniques [17,23,24].

The Schroth Best Practice program consists of Physiologic^®^ exercises, corrections in the activities of daily living, 3D made easy exercises, Schroth exercises, rehabilitation of walk, and de-tethering exercises (Figure 1) [10,17,23].

The clinical exercise program included exercises under the supervision of the same physiotherapist and performed twice a week as a group of two or three patients. General strengthening and stretching exercises were added according to individual needs. This program was also taught to the caregivers of the children, and they were instructed to perform the same program at home on the other days. Performance of the program and progression were checked by experienced physiotherapists. To check compliance with the home exercises, we asked the parents if the exercises were being regularly performed at home. The treatment program was terminated when the Risser grade reached 4. First, the number of exercises was gradually reduced, then the frequency of exercises was reduced to 5 days per week, then to 3. At the end of the treatment, three protective exercises were given as a home program.

“The brace models in the literature regarding the treatment of scoliosis can be listed as Milwaukee, Chêneau, Lyon (ART), Wilmington Boston, TLSO, SPORT and OMC braces. Chêneau based brace models are widely used in worldwide and the most researched brace model in the literature [25]”.

All the patients were fitted with a Chêneau-type brace (Figure 2). The brace was prescribed by an experienced physician and regularly checked. It was recommended to the patients that they wear the brace for 21–23 h per day for the first year. The treatment team asked patients about brace compliance and the maximum daily brace wearing time that was reached was recorded. As the bone maturation reached Risser 4, the daily brace wearing time was gradually reduced by two hours each month. When the daily wearing time was down to 10–12 h, the brace was only worn at night for 6 months, and brace wearing was then terminated.

All the patient data were recorded by the researchers in an Excel program for Windows. The data were then analyzed statistically using SPSS v. 16.00 software. A value of *p* < 0.05 was considered statistically significant for a two-tailed test. Conformity of the data to normal distribution was assessed using the Shapiro–Wilk test. Descriptive statistics were stated as mean ± standard deviation values or number (n) and percentage (%). The paired-samples t-test was used for the analysis of the changes in the Cobb and ATR angles obtained at baseline, at the end of treatment, and in follow-up assessments.

## 3. Results

From a total of 213 patients with scoliosis who presented at the department within the specified period, evaluation was made of 42 who completed the final and follow-up assessments, comprising 38 females and 4 males with a mean age of 11.8 years (see the flow diagram in Figure 3).

The mean total follow-up time was 72.3 months (range 42–125 months) and the mean post-treatment follow-up period was 27.8 months (range 24–71 months).

A total of 25 (59.5%) patients had major thoracic curve, 13 had major lumbar curve, and 4 had thoracolumbar curve. A total of 14 (33.4) patients had flatback. According to the ALS classification, 4C pattern curvature was determined in 15 patients, 3CH in 9, 4CL in 5, 3CN in 5, 3CL in 4, 4CTL in 3, and 3CTL in 1.

The mean Risser grade was 0.2 and the mean Cobb angle of the major curvature was 32.1° (Table 1).

The maximum daily brace-wearing time reached was 20.4 h and the total duration of brace use was 35.1 months (Table 1). A total of 71% of the patients reported a brace-wearing time of ≥20 h per day. Only one patient reported total brace-wearing time of 12 h a day because she did not want to wear it at school. The brace was reported to be worn for 23 h a day by 10 patients.

A statistically significant decrease was determined in both the mean maximum Cobb angle and the maximum ATR value at the end of treatment (*p* < 0.005, Table 2). In the follow-up evaluations, the mean ATR values did not change significantly, while the increase in Cobb angle values was statistically significant (*p* = 0.013, Table 2).

The height of patients increased by an average of 12 cm by the end of the treatment program and by an average of 2 cm from the end of the treatment program to the final follow-up evaluation.

At the end of treatment, the maximum Cobb angle improved in 88.1% of the patients and worsened in 11.9% compared to the baseline values. When the post-treatment and final evaluation results were compared, it was determined that the Cobb angle decreased in one female patient (2.4%), remained stable in 83.3%, and worsened in 14.6%.

The female patient with a decrease of 13° in the major curvature after the treatment had continued to wear the brace for 11 months after the treatment program was terminated and followed the exercise program regularly for 24 months.

The amount of change in the maximum Cobb angle obtained at the end of the treatment, along with age, Risser’s sign, Tanner stage, Cobb angle, and ATR at initial diagnosis, time since menarche, total duration of brace use, and daily brace wearing hours, were analyzed using Spearman’s correlation test. The amount of change in the Cobb angle obtained at the end of treatment was not correlated with other clinical parameters.

## 4. Discussion

The results of this study demonstrated that a conservative treatment program including the Schroth Best Practice program and the Chêneau-type brace, applied with an experienced team approach, is effective in improving the curvature and stopping the progression of adolescent idiopathic scoliosis.

In 2005, the Scoliosis Research Society (SRS) proposed methodological criteria for studies on brace effectiveness [26]. According to the SRS criteria, the inclusion criteria for brace studies were age ≥ 10 years, Risser sign 0–2, Cobb angle 25–40°, no prior treatment, and, if female, less than one year post-menarche at the start of treatment [26]. Patients with a Cobb angle of 20 degrees and above were included in this study. This was because for patients with a Cobb angle below 25 degrees, if the Cobb angle increases by 6 degrees or more between two radiographic evaluations, progression is considered to be present and bracing is recommended [15,17]. Surgical treatment is recommended if the Cobb angle is above 45 degrees in patients whose growth continues, according to the SRS. However, patients and families may want to try conservative treatment first. In the current literature, there are also studies reporting that conservative treatment in cases above 40–45 degrees has been successful [27,28].

Different specific exercise methods and different types of braces for the management of scoliosis have been described in recent literature. Of all the scoliosis-specific exercise approaches, the Schroth method is among the most studied and widely used specific exercise approaches for scoliosis [10,12,23,24]. The Schroth Best Practice program is the latest version of the Schroth method [23]. The 3D Chêneau-type brace is a corrective brace specifically designed for individual curve patterns, and constructed to bring the trunk and spine into the best possible postural and morphological 3D-corrected alignment using a combination of forces applied to the trunk surface [29]. This study used the Schroth Best Practice program and Chêneau-type brace treatment, which are widely used throughout the world. Comparing different types of exercises and braces, and the duration and intensity of treatment, might be specific subjects for future investigations.

Negrini et al. reported that combined bracing and scoliosis-specific exercise in adolescent idiopathic scoliosis based on the SRS and SOSORT criteria is effective and shows better results than those reported in the previous literature, with a 52.3% improvement and 9.6% worsening being reported [13]. In a previous retrospective study by Negrini et al., statistically significant reductions of the scoliosis curvatures > −7.1 were reported for all curves with combined conservative treatment, regardless of the topographic classification [30]. However, patients using different brace models were included in these studies and the exercise program was not standardized.

In a recent study, similar to the current study, either SBP and Chêneau-type brace or Chêneau-type brace treatment were applied. A significantly greater reduction in Cobb angle (−3.5/−0.1) was determined in the group treated with SBP and brace treatment compared to the group receiving only brace treatment. It was also reported that SBP reduced flatback deformity [10]. In this study, the mean Cobb angle was reduced −5.1° after the treatment. At the end of treatment, the maximum Cobb angle had improved in 88.1% of the patients.

Weiss et al. reported that a Gensingen brace (CAD Chêneau-style brace) was successful in 92% of patients with AIS of ≥40° at Risser 0–2 stage [27]. In another study, Weiss et al. applied Chêneau-style Gensingen brace treatment to a patient group with similar characteristics to those of the sample in the current study. By evaluating the radiographs taken after the weaning phase, treatment success was reported to be 80% [11]. In the current study, treatment success was determined to be 85.7% in the evaluation performed on average 24 months after the end of treatment. Although there was an increase (1.9°) in the Cobb angle in the curvatures of some patients at the final evaluation compared to the post-treatment evaluation, the mean values were lower than before treatment. 

Negrini et al. reported ATR improvements of −2.7°, −3.6°, and −3.5° for thoracic, thoracolumbar, and lumbar regions, respectively, after combined treatment in patients at Risser 0–2 with Cobb angle < 40° [30]. Weiss et al. reported 2.1° ATR improvement in the thoracic spine and 1.1° in the lumbar spine in patients with Risser 0–2 and Cobb angle > 40° [27]. In another study, Weiss et al. reported that thoracic ATR reduced 1.8° while the lumbar ATR was reduced 2.7° in patients at Risser 0–2 with Cobb angle < 40° [11]. According to the current study results, the average ATR change was an improvement of −1.7, −3.2, and −1.3 for thoracic, thoracolumbar, and lumbar regions, respectively, after the treatment. As the number of patients was low in some groups, the treatment results were not evaluated according to the pattern or topographic classification of the curve.

The SRS criteria were considered when including patients in the current study. However, the Cobb angles of the patients included ranged between 20° and 47°, with six patients having a maximum Cobb angle of >40°. The mean Cobb angle was 43.6° at the beginning of the study, 36.3° at the end of treatment, and 39.8° at the follow-up evaluation. In these patients with high growth potential, and therefore high risk of progression, achieving even stabilization in long-term follow-up can be considered a success. The SRS criteria were published in 2005 and since then many studies have been conducted and knowledge has increased. Previous studies have shown curvature progression in children with high growth potential who were only observed [6,12]. According to the SRS, surgical treatment is recommended for adolescent idiopathic scoliosis with a maximum Cobb angle > 45°. However, in many European countries, conservative treatment is the priority [10,23]. Even if surgical treatment is recommended, families and their children may wish to try conservative treatment. Therefore, in future studies in which brace treatment will be applied, the inclusion criteria may need to be reconsidered.

Since one-third of the iliac apophysis cannot be seen on the frontal radiograph, Kotwicki claimed that the Risser sign grading method currently in use does not consider the actual excursion of the iliac apophysis [31]. In clinical practice, curve magnitude and Risser evaluations are performed on AP radiographs. In addition, due to reports of residual growth following full iliac apophysis excursion (Risser 4), several authors [32,33] have questioned the Risser sign’s ability to predict the cessation of spinal growth. Little and Sussman [32] came to the conclusion that the Risser sign is not reliable enough to replace a hand–wrist radiograph as a gauge of skeletal maturity in a particular patient. The full excursion of the iliac crest (Risser 4), according to the analysis of Hoppenfeld et al., is not a definitive sign that spinal growth has come to a halt [34]. Little et al. [32] found that 10% of patients have maximum curve progression after Risser grade 4, and Kotwicki demonstrated that 8% of Risser 4 patients did not reveal complete excursion [31]. In the current study, it was determined that some patients had an increase in curvature after the treatment was terminated. The height of those patients increased by an average of 2.3 cm (n = 6) after treatment, whereas those whose curvature remained stable increased by an average of 1.4 cm (n = 36). This difference was not significant, but in future studies different evaluation methods for maturation can be compared as treatment outcome criteria. Accordingly, the treatment period may be slightly longer in patients with high growth potential.

Although there has also been an increase in research and evidence about the conservative treatment of scoliosis, studies have mostly investigated the effectiveness of brace use alone, and very few have included both methods [10,13,30,35]. In order to improve treatment outcomes [10,13], prevent corrective loss following brace weaning [36], and dramatically lower the likelihood of surgery [35], patients should preferably practice scoliosis-specific exercises in addition to brace therapy. Our clinical observations also show that less improvement is obtained in adolescents using a brace without exercising, or vice versa. In addition, the performance of group exercises might improve compliance with the continuity of exercise and brace use. Including the combination of exercise and brace, an experienced team approach, the application of the same type of brace for all the patients, and performing and progressing the exercises with the same physiotherapist can be considered strengths of this study. The follow-up period of at least 24 months after treatment can be considered as the strongest aspect of the study, as previous studies in the literature have generally only presented the results obtained at the end of treatment.

There were some shortcomings in this study, primarily that the braces did not include any sensors to monitor wear time, so compliance was evaluated only by questioning the patients and parents. A second limitation was the different numbers of patients in the pattern groups for the analysis of treatment outcomes according to curve pattern. The fact that the study did not have a control group, and that the treatment applied could not be compared with different treatment methods, can be considered as other limitations of the study. Other outcome measurements were not included, such as quality of life or brace stress. Finally, maturation was not evaluated with hand radiography for all patients.

## 5. Conclusions

It is very important to apply the right treatment at the right time in growing children with adolescent idiopathic scoliosis. Wasting time with inappropriate treatments leads to an increase in the curvature, an increase in the need for surgical treatment, and financial and psychological problems for the family and the patient. Therefore, treatment methods should be applied in the light of evidence-based information by a professional healthcare team without wasting time. Our clinical experience shows that good and regular communication with the patient and family increases adherence to treatment. Patients with AIS who receive combined exercise and brace treatment have increased adherence to treatment compared to other patients receiving only treatment with one or the other. With proper conservative management in cooperation with the patient, family, and clinician, it is possible to correct the curvature and stop the progression to a great extent. It can be recommended that clinicians and researchers consider these points in the conservative management of AIS.

## Figures and Tables

**Figure 1 children-10-00386-f001:**
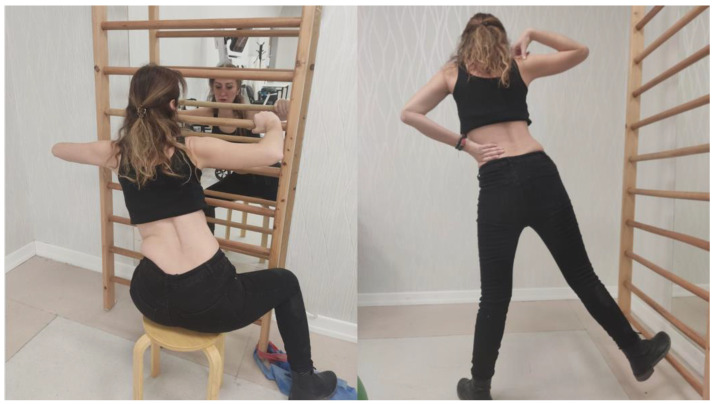
Examples of Schroth Best Practice exercises.

**Figure 2 children-10-00386-f002:**
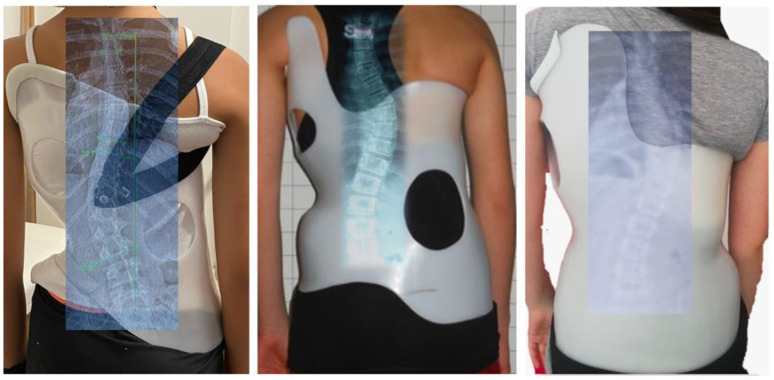
Examples of Chêneau-type braces.

**Figure 3 children-10-00386-f003:**
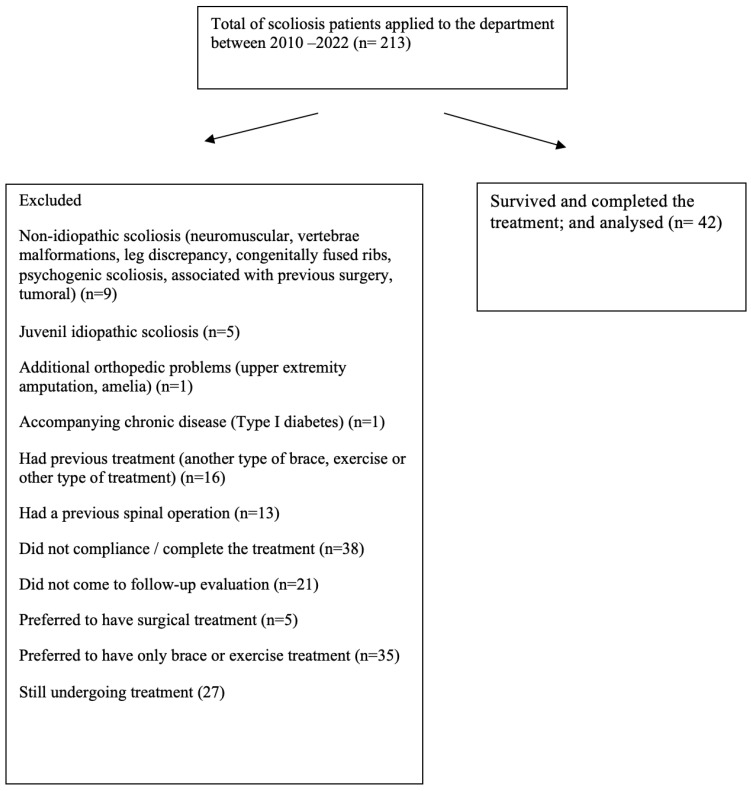
Flow diagram of the participants.

**Table 1 children-10-00386-t001:** Clinical characteristics of the patients.

Variables	Mean ± SD Median (min–max)
Age (years)	11.8 ± 1.3 11 (10–14.5)
Risser grade	0.2 ± 0.5 0 (0–2)
Tanner stage	1.9 ± 1 2 (0–4)
Height (cm)	153.7 ± 11.4 (121–175)
Weight (kg)	42.5 ± 9.6 (23–62)
BMI (kg/m^2^)	18.1 ± 2.8 (12.7–23.6)
Reached maximum daily brace wearing time (hours)	20.4 ± 2.4 (12–23)
Total duration of brace use (months)	35.1 ± 17.4 29.5 (18–87)

**Table 2 children-10-00386-t002:** Change in outcome measurements over time.

	Start of the Treatment Mean ± SD (min–max)	End of the Treatment Mean ± SD (min–max)	Last Follow-Up Mean ± SD (min–max)	*p* Value	Mean Change
**Cobb angle**	32.1 ± 7.3 (20–47)	26.9 ± 9.9 (10–48)	28.8 ± 10.1 (12–48)	* 0.000	C1 −5.1 ± 7.3 (−17–12)
				** 0.013	C2 −3 ± 7.4 (−17–15)
				*** 0.007	C3 1.9 ± 4.7 (−13–18)
**ATR**	8.2 ± 3.7 (2–17)	6.2 ± 3.1 (1–13)	6.8 ± 3 (1–16)	* 0.001	C1 −1.7 ± 3.6 (−11–6)
				** 0.113	C2 −1.1 ± 3.4 (−8–6)
				*** 0.014	C3 0.6 ± 2.5 (−6–9)

* Comparison of angle at the start of the treatment and end of the treatment; ** comparison of the angle at the end of the treatment and at the last follow-up; *** comparison of the angle at the start of the treatment and at the last follow-up. C1: Mean change from start of the treatment to the end of the treatment; C2: mean change from start of the treatment to the last follow-up; C3: mean change from end of the treatment to the last follow-up.

## Data Availability

The data presented in this study are available on request from the corresponding author. The data are not publicly available due to data collection is ongoing.

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
