# Peer review of "The Effectiveness of the Schroth Best Practice Program and Chêneau-Type Brace Treatment in Adolescent Idiopathic Scoliosis: Long-Term Follow-Up Evaluation Results"

_children, 2023, doi:10.3390/children10020386_

Round 1
Reviewer 1 Report
I appreciated the paper by Colak and colleagues about using the Schroth best practice program and bracing in AIS patients. Finding new conservative solutions for AIS patients is mandatory to avoid surgery.
However, the study has some methodological issues and limitations.
Firstly, a native English speaker should revise the paper as there are some grammatical errors and several sentences are challenging to read. Moreover, there are some errors in the text (some sentences are reported in a different font).
The authors found their results on the basis that any improvements in cobb value are significant. However, in the international literature, a meaningful improvement in scoliosis is defined when >5° is reached. This value is higher than those reported by the authors.
Why did the authors use the ALS classification? First, the Lenke classification is the most reliable and known classification for AIS patients. Moreover, the compliance of the patients with bracing was not assessed.
Table 1 has some grammatical errors (orhopadic) and layout issues).
Providing significant results about the use of bracing is challenging, as Weinstein reported in his most famous study. Moreover, I also have some other minor comments.
One potential weakness of this study is that it is retrospective, which may introduce bias in the selection of patients and data collection. Additionally, the study may not have a control group, which makes it difficult to compare the effectiveness of the treatment to a standard of care or a different treatment method. The sample size may also be considered negligible, as only a limited number of patients were included in the study. The study also lacks information about the long-term compliance of the patient to the treatment and the adherence to the recommended exercises or bracing, which can affect the outcome of the treatment.
Author Response
Dear editor and reviewers,
We thank you for your careful reading of our manuscript and thoughtful comments. Below, we have addressed your questions and comments and indicated the changes in the manuscript (highlighted in green in the revised text).
Reviewer 1:
I appreciated the paper by Colak and colleagues about using the Schroth best practice program and bracing in AIS patients. Finding new conservative solutions for AIS patients is mandatory to avoid surgery.
However, the study has some methodological issues and limitations.
- Firstly, a native English speaker should revise the paper as there are some grammatical errors and several sentences are challenging to read. Moreover, there are some errors in the text (some sentences are reported in a different font).
Response:
Dear reviewer, as you suggested, our manuscript was checked by a native English speaker.
- The authors found their results on the basis that any improvements in cobb value are significant. However, in the international literature, a meaningful improvement in scoliosis is defined when >5° is reached. This value is higher than those reported by the authors.
Response:
In the method section, we have also explained that “The difference between the Cobb angle in the follow-up examinations and before treatment was calculated. Based on this analysis, according to the SOSORT (International society on scoliosis orthopaedic and rehabilitation treatment) guidelines, three possible outcomes are categorised: curve correction (>-5°), curve stabilization (>-5° and <5°), and curve progression (>5°) [16].”
And we have given the results according to this categorisation in the results section.
“At the end of treatment, the maximum Cobb angle improved in 88.1% of the patients and worsened in 11.9% compared to the baseline values. When the post-treatment and final evaluation results were compared, it was determined that the Cobb angle decreased in 1 female patient (2.4%), remained stable in 83.3%, and worsened in 14.6%. “
- Why did the authors use the ALS classification? First, the Lenke classification is the most reliable and known classification for AIS patients. Moreover, the compliance of the patients with bracing was not assessed.
Response:
Lenke classification is mostly used by surgeons in surgical treatment applications.
Ref 1: Slattery C, Verma K. Classifications in Brief: The Lenke Classification for Adolescent Idiopathic Scoliosis. Clin Orthop Relat Res. 2018 Nov;476(11):2271-2276. doi: 10.1097/CORR.0000000000000405.
Ref 2: Lenke LG, Edwards CC, Bridwell KH. The Lenke classification of adolescent idiopathic scoliosis: how it organizes curve patterns as a template to perform selective fusions of the spine. Spine (Phila Pa 1976). 2003;28:S199–207.
Schroth, ALS and Rigo classifications are widely used in conservative treatment of scoliosis.
The individuals included in this study were treated with Schroth Best Practice program and bracing. For the exercises applied in the Schroth Best Practice program, the curvature patterns of the patients should be defined according to the ALS classification. Exercises are applied to patients according to these specific patterns.
Previous studies have demonstrated the validity and reliability of the ALS classification.
Ref 3: Akçay, B.; Çolak, T. K.; Apti, A.; Çolak, İ.; KızıltaÅŸ, Ö. The reliability of the augmented Lehnert-Schroth and Rigo classification in scoliosis management. The South African journal of physiotherapy 2021, 77(2), 1568. https://doi.org/10.4102/sajp.v77i2.1568
- Table 1 has some grammatical errors (orhopadic) and layout issues).
Response: We have revised.
- Providing significant results about the use of bracing is challenging, as Weinstein reported in his most famous study. Moreover, I also have some other minor comments.
One potential weakness of this study is that it is retrospective, which may introduce bias in the selection of patients and data collection.
Response: Dear referee, we would like to thank you for your valuable comment. We will take your suggestion into consideration in our future studies.
However, in reality, we took care to include all children who applied to our department and had long-term follow-up.
- Additionally, the study may not have a control group, which makes it difficult to compare the effectiveness of the treatment to a standard of care or a different treatment method.
Response: We added your suggeestion to our manuscript as a limitation.
“The fact that the study did not have a control group or that the treatment applied could not be compared with different treatment methods can be considered as another limitation of the study.”
- The sample size may also be considered negligible, as only a limited number of patients were included in the study.
Response: Dear reviewer, when we calculate the effect size as 0.5 and power as 90% with the GPower 3.1 program, the number of patients who should be included in our study is 36.
However, we included 42 patients in our study. and the number of studies that follow their AIS patients for 2 years after treatment and present their results is quite limited.
- The study also lacks information about the long-term compliance of the patient to the treatment and the adherence to the recommended exercises or bracing, which can affect the outcome of the treatment.
Response:
“The clinical exercise program included exercises under the supervision of the same physiotherapist performed twice a week as a group of 2 or 3 patients. This program was also taught to the caregivers of the children, and they were instructed to perform the same program at home on the other days. The treatment program was terminated when the Risser grade was 4.”
Therefore, until the risser 4 stage, children exercised under the supervision of a physiotherapist. It was checked by asking the families whether the exercises were performed regularly at home.
We added:
“To check the compliance to home exercises, we asked the parents if the exercises were regularly performed at home”
For brace compliance, “Brace compliance was questioned by the treatment team and the maximum daily brace wearing time that was reached was recorded.”

Reviewer 2 Report
· The format of the manuscript is not consistent (such as fonts in page 2, line 43 & line 64 & line 84).
· Page 2, line 75: The inclusion criteria were “Cobb angle > 20º”. Page 5, table 2: The result showed that at the beginning, the Cobb angles of the patients were 20º -47º. According to the SRS and SOSORT criteria mentioned above in this manuscript, the inclusion criteria for brace studies should be Cobb angle 25-40°. Please clarify the inclusion criteria of the Cobb angle of the patients with AIS.
· Page 3, line 127-132: Was the twice-a-week supervised exercise program ongoing throughout the whole treatment period? How was the compliance with home exercise monitored and recorded?
· Page 5, line 161: The flatback was mentioned in the result, but there was no relevant background information, and more information should be provided about the flatback situation in the patient with AIS.
· Page 5, line 168-172: In the result, the range of the brace-wearing time was from 12 hours/day to 23 hours/day. As mentioned before (page 4), the patients did not comply with the treatment were excluded. Please clarify the exclusion criteria of the compliance of the brace-wearing time. The analyses should also include the concept of intention-to-treat in order to reflect the actual clinical situation of the patients and the relevant treatment outcome.
· This study focused on the treatment results of combined spinal orthosis and exercise while there was no control group for comparison.
Author Response
Dear editor and reviewers,
We thank you for your careful reading of our manuscript and thoughtful comments. Below, we have addressed your questions and comments and indicated the changes in the manuscript (highlighted in green in the revised text).
Reviewer 2:
- The format of the manuscript is not consistent (such as fonts in page 2, line 43 & line 64 & line 84).
Response: We have checked and corrected.
- Page 2, line 75: The inclusion criteria were “Cobb angle > 20º”. Page 5, table 2: The result showed that at the beginning, the Cobb angles of the patients were 20º -47º. According to the SRS and SOSORT criteria mentioned above in this manuscript, the inclusion criteria for brace studies should be Cobb angle 25-40°. Please clarify the inclusion criteria of the Cobb angle of the patients with AIS.
Response: Dear referee, we also agree with your comment. We have deleted the paragraph in the introduction section in line with your suggestion and clearly stated our inclusion criteria.
However, for children with a Cobb angle of less than 25 degrees but whose Cobb angle increases by 6 degrees or more during two radiographic evaluations, bracing may be recommended, anticipating that the curve will progress.
Conservative treatment is also a treatment option for families and children with curvatures over 40 degrees who want to try conservative treatment first.
In the current literature, there are also studies reporting that conservative treatment above 40 - 45 degrees results in success.
Ref 1: Aulisa AG, Guzzanti V, Falciglia F, Giordano M, Galli M, Aulisa L. Brace treatment of Idiopathic Scoliosis is effective for a curve over 40 degrees, but is the evaluation of Cobb angle the only parameter for the indication of treatment? Eur J Phys Rehabil Med. 2019 Apr;55(2):231-240. doi: 10.23736/S1973-9087.18.04782-2
Ref 2: Weiss HR, Tournavitis N, Seibel S, Kleban A. A Prospective Cohort Study of AIS Patients with 40° and More Treated with a Gensingen Brace (GBW): Preliminary Results. Open Orthop J. 2017 Dec 29;11:1558-1567. doi: 10.2174/1874325001711011558.
- Page 3, line 127-132: Was the twice-a-week supervised exercise program ongoing throughout the whole treatment period? How was the compliance with home exercise monitored and recorded?
Response:
“The clinical exercise program included exercises under the supervision of the same physiotherapist performed twice a week as a group of 2 or 3 patients. This program was also taught to the caregivers of the children, and they were instructed to perform the same program at home on the other days. The treatment program was terminated when the Risser grade was 4.”
Therefore, until the risser 4 stage, children exercised under the supervision of a physiotherapist. It was checked by asking the families whether the exercises were performed regularly at home.
We added:
“To check the compliance to home exercises, we asked the parents if the exercises were regularly performed at home”
- Page 5, line 161: The flatback was mentioned in the result, but there was no relevant background information, and more information should be provided about the flatback situation in the patient with AIS.
Response: We have added “Sagittal plane evaluation was performed on lateral radiographs with Cobb method.” in the method section.
- Page 5, line 168-172: In the result, the range of the brace-wearing time was from 12 hours/day to 23 hours/day. As mentioned before (page 4), the patients did not comply with the treatment were excluded. Please clarify the exclusion criteria of the compliance of the brace-wearing time. The analyses should also include the concept of intention-to-treat in order to reflect the actual clinical situation of the patients and the relevant treatment outcome.
Response: Dear editor, the daily wearing time of the brace is not specified as an inclusion period.
Only one patient reported total brace-wearing time of 12 hours a day because she did not want to wear it at school.
The inclusion criteria were defined as a diagnosis of adolescent idiopathic scoliosis, age ≥10 years, Cobb angle≥ 20º, no other treatment which might affect scoliosis, and follow up of at least 24 months after the end of treatment. The exclusion criteria were the presence of any contraindications to exercise or brace treatment, accompanying mental or pyschological problems, any chronic, neurological- muscular or rheumatic diseases, any previous or ongoing treatment, accompanying orthopaedic problems, non-idiopathic scoliosis, or refusal of the recommended treatment.
To present the actual results of treatment, the authors did not include intention- to- treat analysis in this study.
- This study focused on the treatment results of combined spinal orthosis and exercise while there was no control group for comparison.
Response: We added your suggeestion to our manuscript as a limitation.
“The fact that the study did not have a control group or that the treatment applied could not be compared with different treatment methods can be considered as another limitation of the study.”
However, previous studies have reported progression of curves in control groups left untreated.
Ref 1: Kuru, T.; Yeldan, İ.; Dereli, E. E.; Özdinçler, A. R.; Dikici, F.; Çolak, İ. The efficacy of three-dimensional Schroth exercises in adolescent idiopathic scoliosis: a randomised controlled clinical trial. Clinical rehabilitation 2016, 30(2), 181–190. https://doi.org/10.1177/0269215515575745

Reviewer 3 Report
The paper is interesting and relevant and may be published, but please note the following comments:
1 - The authors can mention in the introduction something about the evolution of the use of vests, remembering the use of the Milwaukee and OTLS vest, with which treatment protocols were developed, reporting the limitations and advantages of the current treatment.
2 - It would be interesting if the authors included some figures in the paper, illustrating the radiological evolution of a patient, showing the use of the vest and, if possible, performing exercises.
3- The big problem with the paper is that two techniques were used that certainly contributed to the good results!
In the authors' opinion, what was more important for the result, the exercises or the use of the vest?
Logically, the ideal would be for there to be a comparison with 1 group that only used a vest, or only performed the exercise technique. Even so, the data presented are relevant, they can be published, but I ask that the authors give more prominence and discuss this problem better!
4- The conclusion is extensive and contains facts that are not in the study proposal and should not be in this topic, but in the discussion!
Authors should only report the result obtained using the exercise program and vest. There are no data that show, although it is obvious, that a good relationship and communication with the patient and family helps in the results. This is an observation based on the authors' previous experience and should not be included in the conclusion!
In addition, no comparative study was carried out between groups that exercised and used vests compared to the group that only wore vests.
Author Response
Dear editor and reviewers,
We thank you for your careful reading of our manuscript and thoughtful comments. Below, we have addressed your questions and comments and indicated the changes in the manuscript (highlighted in green in the revised text).
Reviewer 3:
The paper is interesting and relevant and may be published, but please note the following comments:
1 - The authors can mention in the introduction something about the evolution of the use of vests, remembering the use of the Milwaukee and OTLS vest, with which treatment protocols were developed, reporting the limitations and advantages of the current treatment.
Response:
We added in the method section,
“The brace models in the literature regarding the treatment of scoliosis can be listed as Milwaukee, Chêneau, Lyon (ART), Wilmington Boston, TLSO, SPORT and OMC braces. Chêneau based brace models are widely used in worldwide and Chêneau brace is one of the most researched brace model in the literature “
2 - It would be interesting if the authors included some figures in the paper, illustrating the radiological evolution of a patient, showing the use of the vest and, if possible, performing exercises.
Response:
3- The big problem with the paper is that two techniques were used that certainly contributed to the good results!
In the authors' opinion, what was more important for the result, the exercises or the use of the vest?
Logically, the ideal would be for there to be a comparison with 1 group that only used a vest, or only performed the exercise technique. Even so, the data presented are relevant, they can be published, but I ask that the authors give more prominence and discuss this problem better!
The conclusion is extensive and contains facts that are not in the study proposal and should not be in this topic, but in the discussion!
Authors should only report the result obtained using the exercise program and vest. There are no data that show, although it is obvious, that a good relationship and communication with the patient and family helps in the results. This is an observation based on the authors' previous experience and should not be included in the conclusion!
In addition, no comparative study was carried out between groups that exercised and used vests compared to the group that only wore vests.
Response:
Dear editor, we have answered your comments item by item. The importance of our study is that it includes long-term follow-up for this comprehensive conservative treatment method.
- In the discussion section, we had stated that
“Our clinical observations also show that less improvement is obtained in adolescents using a brace without exercising or vice versa.”
- The authors only mentioned this experience and their observations.
- In addition, appropriate indications for the treatment of scoliosis are clearly stated in the literature in important guidelines. Cobb angle, Risser's sign and age are important parameters in determining these indications. If the patient has an indication for bracing and exercise according to the calculated progression risk factor and if only exercise is performed, it is obvious that the curve will progress.
Ref 1: Lonstein JE, Carlson JM. The prediction of curve progression in untreated idiopathic scoliosis during growth. J Bone Joint Surg Am. 1984;66(7):1061-1071.
Ref 2: Dereli EE, Gong S, Çolak TK, Turnbull D. Guidelines for the conservative treatment of spinal deformities - Questionnaire for a Delphi consensus. S Afr J Physiother. 2021 Dec 10;77(2):1587. doi: 10.4102/sajp.v77i2.1587.
Ref 3: SOSORT guideline committee; Weiss HR, Negrini S, Rigo M, Kotwicki T, Hawes MC, Grivas TB, Maruyama T, Landauer F. Indications for conservative management of scoliosis (guidelines). Scoliosis. 2006 May 8;1:5. doi: 10.1186/1748-7161-1-5.
Ref 4: Weiss, H. R.; Weiss, G.; Schaar, H. J. Conservative management in patients with scoliosis--does it reduce the incidence of surgery?. Studies in health technology and informatics 2002, 91, 342–347.
- It has also been reported in the literature that the addition of Schroth exercises to brace treatment increases the success of treatment.
Ref 5: Fang MQ, Huang XL, Wang W, Li YA, Xiang GH, Yan GK, Ke CR, Mao CH, Wu ZY, Pan TL, Zhu RB, Xiao J, Yi XH. The efficacy of Schroth exercises combined with the Chêneau brace for the treatment of adolescent idiopathic scoliosis: a retrospective controlled study. Disabil Rehabil. 2022 Sep;44(18):5060-5068. doi: 10.1080/09638288.2021.1922521.
- Exercise and brace therapy was indicated for all patients included in this study. Patients are advised to take both treatments together, but sometimes patients do not comply despite proper information.
- As a study design, the initial Cobb angles of the patients must be different so that we can recommend only exercise to a group and only brace treatment to another group. Patients who will only be offered exercise therapy should have mild Cobb angles and low progression risks. Therefore, it will not be possible to compare the two groups homogeneously.

Round 2
Reviewer 1 Report
The authors improved the paper